DISCOVERY REPORT

# Short-range human cortico-cortical white matter fibers have thinner axons and are less myelinated compared to long-range fibers despite a similar g-ratio

Philip Ruthig[1][ꙩ]*, David Edler von der Planitz[1][ꙩ], Maria Morozova[1,2], Katja Reimann[1], Carsten Jäger[1,2], Tilo Reinert[1], Siawoosh Mohammadi[2,3,4,5], Nikolaus Weiskopf[2,6,7], Evgeniya Kirilina[2], Markus Morawski[1,2]*

1 Paul Flechsig Institute—Centre of Neuropathology and Brain Research, Medical Faculty, University of Leipzig, Leipzig, Germany, 2 Department of Neurophysics, Max Planck Institute for Human Cognitive and Brain Science, Leipzig, Germany, 3 Department of Neuroradiology, University of Lübeck, Lübeck, Germany, 4 Institute of Systems Neuroscience, University Medical Center Hamburg-Eppendorf, Hamburg, Germany, 5 Max Planck Research Group MR Physics, Max Planck Institute for Human Development, Berlin, Germany, 6 Felix Bloch Institute for Solid State Physics, Faculty of Physics and Earth System Sciences, Leipzig University, Leipzig, Germany, 7 Wellcome Centre for Human Neuroimaging, Institute of Neurology, University College London, London, United Kingdom

ꙩ These authors contributed equally to this work.
* philip.ruthig@gmail.com (PR); Markus.Morawski@medizin.uni-leipzig.de (MM)

The Editors encourage authors to publish research updates to this article type. Please follow the link in the citation below to view any related articles.

## Abstract

The size and complexity of the human brain require optimally sized and myelinated fibers. White matter fibers facilitate fast communication between distant areas, but also connect adjacent cortical regions via short association fibers. The difference in length and packing density of long and short association fibers pose different requirements on their optimal size and degree of myelination. The fundamental questions of (i) how thick the short association fibers are and (ii) how strongly they are myelinated as compared to long fibers, however, remain unanswered. We present a comprehensive two-dimensional transmission electron microscopic analysis of ~400,000 fibers of human white matter regions with long (corpus callosum) and short fibers (superficial white matter). Using a deep learning approach, we demonstrate a substantially higher fiber diameter and higher myelination thickness (both approximately 25% higher) in corpus callosum than in superficial white matter. Surprisingly, we do not find a difference in the ratio between axon diameter and myelin thickness (g-ratio), which is close to the theoretically optimal value of ~0.6 in both areas (0.54). This work reveals a fundamental principle of brain organization that provides a key foundation for understanding the human brain.

**Data availability statement:** All of the raw and processed data is available upon request. All text- and image-based data (csv files with axon & outer fiber diameters, g-ratio measurements) underlying figures Figs 2, 3, 4, S1-11, and scripts that process these data are available at github.com/PhilipRuthig/ShortvsLongfibers/ (https://doi.org/10.5281/zenodo.15601201). The repository also contains sample image data of CC and SWM at every step in the pipeline. All text-based data is available through a Zenodo archive (https://zenodo.org/records/15720452). All of the code used for the analyses in this study is publicly available on Github (github.com/PhilipRuthig/ShortvsLongfibers/, archived release version: Zenodo. https://doi.org/10.5281/ZENODO.15601202). The deep learning model and accompanying training data will be published at a later date, but are available upon request.

**Funding:** MM, SM, NW, and EK, were supported by the German Research Foundation (DFG Priority Program 2041 'Computational Connectomics' MO 2249/3-1, MO 2249/3-2, KI 1337/2-2, WE 5046/4-2, MO 2397/5-1, MO 2397/5-2; https://www.dfg.de/de/aktuelles/neuigkeiten-themen/info-wissenschaft/2020/info-wissenschaft-20-19). SM was additionally supported by the DFG Emmy Noether program MO 2397/4-1 and MO 2397/4-2 (https://www.dfg.de/de/foerderung/foerdermoeglichkeiten/programme/einzelfoerderung/emmy-no-ether) and the ERC (Acronym: MRStain, Grant agreement ID: 101089218, https://doi.org/10.3030/101089218; https://erc.europa.eu/homepage). Herewith, we declare that the funders had no role in study design, data collection and analysis, decision to publish, or preparation of the manuscript.

**Abbreviations:** CC, corpus callosum; GEV, Generalized Extreme Value; MCMC, Markov chain Monte Carlo; SWM, superficial white matter.

## Introduction

Myelinated fibers in the white matter connect both nearby and distant cortical areas. These connections are crucial for various functions, including sensory-motor integration [1,2], language perception [3], and inter-hemispheric information transfer [4]. At the same time, white matter fibers can be highly plastic, and this plasticity is behaviorally relevant, e.g., during learning [5]. Reflecting this functional diversity, white matter fibers can be highly variable in length and diameter [6]. Long-range projections include corticospinal tracts, which can extend up to 1 m length [7,8], and, e.g., projections to the prefrontal cortex [9]. In contrast, short-range connections span only a few centimeters and link adjacent areas such as the motor and somatosensory cortex [1,10,11], different visual cortical areas [12–14], or auditory cortical fields [15].

Axon diameter and myelination thickness are key factors influencing the conduction velocity of a fiber. At the same time, these factors also define a fiber's energy demand and the space it occupies: the higher in diameter an axon and the thicker its myelination, the faster it can propagate signals between distant regions of the human brain. However, increasing fiber diameter and myelination thickness also increases the metabolic cost dramatically—even when the fiber is not actively conducting action potentials [16,17]. Consequently, large-diameter and heavily myelinated axons are rare [18]. Theoretically, a fiber optimized for maximum conduction velocity in the CNS has a g-ratio (ratio of axon diameter to outer fiber diameter) of 0.6–0.77 [19,20].

From an evolutionary perspective, the trade-off for thicker and more myelinated fibers is justified only when rapid and efficient signal transmission is functionally required. Such functions may include rapid integration of somatosensory feedback to the motor cortex and vice versa [11], relaying temporally complex auditory cues to higher-order areas [21–23], or rapid conduction of visual stimuli via the optic nerve to the visual cortex [24]. A long-standing hypothesis posits that longer fiber tracts are associated with larger axon diameters and increased myelination, facilitating more efficient and rapid signal conduction over extended distances [25–28]. Although widely accepted, this hypothesis still lacks ultrastructural, gold standard evidence based on the human brain.

In this study, we investigate this fundamental principle of white matter fiber organization. Specifically, we analyze the axon diameter and myelination thickness in regions with either predominantly short or predominantly long-range fiber connections. We compared two sets of regions using two-dimensional transmission electron microscopy: (i) long range fibers sampled from the entire corpus callosum (CC), and (ii) short range fibers sampled from superficial white matter (SWM, sampled <3 mm from the white/gray matter border) between primary motor (M1) and somatosensory cortex (S1), and between primary (V1) and secondary (V2) visual cortex (Fig 1). In SWM regions, we expect a higher proportion of short association fibers ('U-fibers') connecting M1-S1 and V1-V2. These short fiber tracts have previously been described using gross anatomy or diffusion MRI, showing perpendicular orientation to the central sulcus [1,10,11], and in the visual cortex [12–14,29]. However, a detailed ultrastructural description has long been lacking.

PLOS Biology

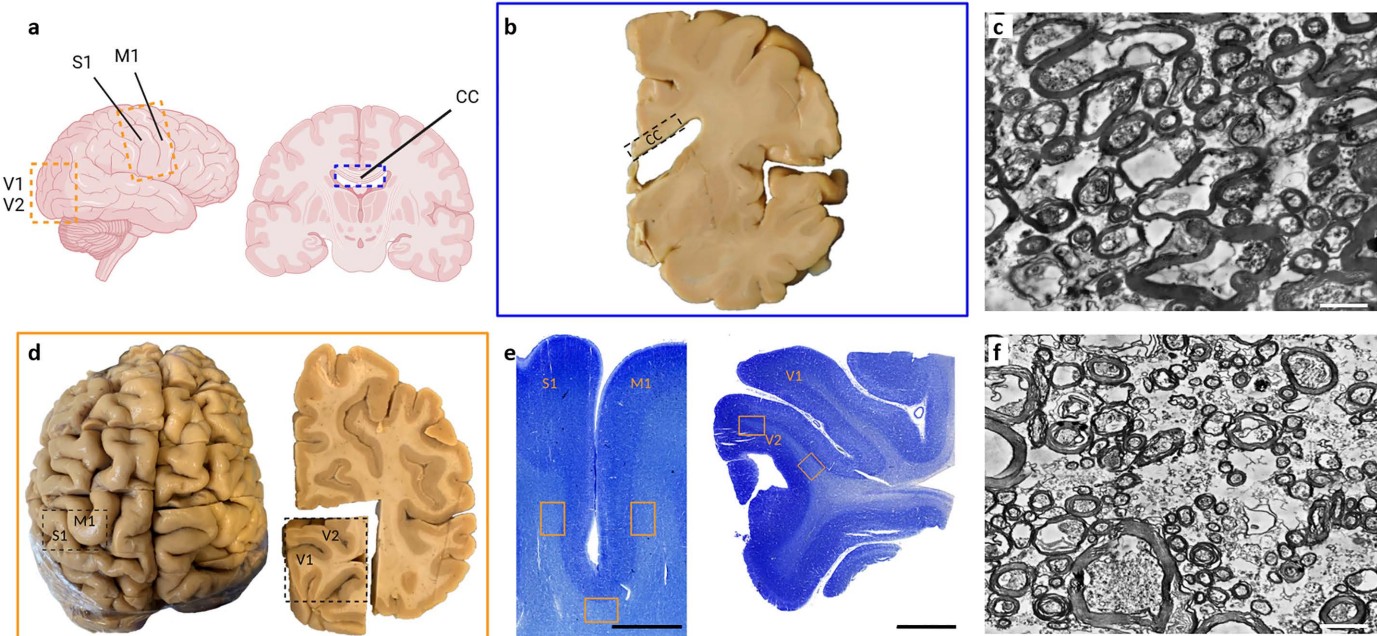

**Fig 1. Overview over brain regions sampled in this study.** a, Depiction of a human brain and brain slice with the used superficial white matter (SWM) regions (yellow) and corpus callosum (CC) regions (blue) highlighted. b, Gross anatomical view of a coronal slice, containing CC. c, Transmission electron micrograph of CC tissue. d, gross anatomical location of brain regions used for SWM tissue sampling. e, Nissl stainings of tissue containing SWM along the central sulcus (left) with short association fibers connecting S1 and M1 and short association fibers connecting primary (V1) and secondary (V2) visual cortices (right), with example regions that were studied highlighted (yellow). f, Transmission electron micrograph of SWM tissue. Scale bars in e are 5 mm. Scale bars in c and f are 2 µm. Data available in S1 Data.

The most comparable study to date was published very recently and quantifies the ultrastructure of myelinated fibers in SWM [30]. However, in this study, no comparison to the CC was conducted. Another relevant study on fetal sheep brain tissue found that the SWM contains 50% more thin axons (diameter <0.65 µm) than the CC, and slightly more thick (diameter >8.5 µm) axons [31]. Most previous studies of the CC have either employed relatively distant model organisms or relied on exceedingly small fields of views and numbers of manually segmented fibers [31–33]. The development of deep learning segmentation algorithms now enables the analysis of far larger fiber populations.

In the present study, we employ a state-of-the-art automated segmentation algorithm to analyze large quantities of high-quality human tissue samples. In this study, we show that short association fibers have smaller axon and outer fiber diameters compared to long-range fibers. We provide experimental evidence for the distinct structural characteristics of short association fibers in the human brain and highlight the tight link between function and structure in different white matter fiber tracts.

## Results

### Short-range fibers have smaller axon diameters and thinner myelin compared to long-range fibers

To understand systematic structural differences between long and short fibers, we assessed the axon diameter (here assumed to be identical with the inner myelin diameter) and the outer fiber diameter (i.e., axon+myelination diameter) in SWM and CC. In contrast to previous studies, which frequently sampled only small numbers of fibers, we automatically segmented and analyzed approximately 200,000 fibers in CC and SWM each.

We studied samples from five donor brains (4 male, 1 female, aged 60–78 years) and generated two-dimensional transmission electron micrographs of both CC and SWM (Fig 2a and 2d). To reliably segment myelinated fibers, we trained a DenseNet [34], which produces semantic predictions of axons, myelin, and background (Fig 2b and 2e). This preliminary segmentation was then processed into a final segmentation of axon + myelin pairs (Fig 2c and 2f), which were subsequently measured and analyzed (Fig 2g–2i).

We show that SWM axons are systematically thinner (median axon diameter: 0.61 µm (SWM) versus 0.76 µm (CC)) and less myelinated than fibers in CC (median outer fiber diameter: 0.94 µm (SWM) versus 1.19 µm (CC), see Fig 3a and 3b). We described these distributions by fitting a Generalized Extreme Value (GEV) function to each distribution of axon diameters and outer fiber diameters, using Markov chain Monte Carlo (MCMC) sampling (Figs 3a, 3b, 4 and S10–S12). Each GEV distribution is defined by three parameters: the location parameter $\mu$ (reflecting the location of the center of mass in the distribution of radii), the scale parameter $\sigma$ (defining the width of the distribution), and the shape parameter $\xi$ (defining the tailedness of the distribution). These parameters are shown in Fig 4b, demonstrating that the distributions of axon and outer fiber diameters in CC are similar in shape to those in SWM, but are broader and shifted towards larger diameters. GEV distributions for CC axons and outer fiber diameters show higher $\mu$ and $\sigma$ values than the corresponding SWM distributions, while shape parameters are comparable. The maxima of the distributions are 0.38 µm for SWM axon diameters, 0.76 µm for SWM outer fiber diameters, 0.5 µm for CC axon diameters, and 0.98 µm for CC outer fiber diameters.

### Superficial white matter fibers show greater structural diversity than corpus callosum fibers

This raises the question if this difference is also reflected in the g-ratio? Although CC and SWM differ markedly in axon diameter and outer fiber diameter (Fig 3a and 3b), we found that the mean g-ratio is virtually identical between these groups (SWM: 0.538, CC: 0.537, Fig 3e). However, the g-ratio distributions reveal a slightly higher proportion of weakly myelinated fibers in SWM compared to CC. When analyzing the relationship between axon diameter and g-ratio (Fig 3c and 3d), we observed that larger fibers in both CC and SWM tend to cluster around g-ratio = 0.6.

### Region-specific anatomical specialization might underlie conduction velocity differences

Finally, the important question arises: What are the functional implications of the observed anatomical differences? Although various factors influence the conduction velocity of a fiber (e.g., the ratio of the internode-to-node length), it can be reasonably estimated from the axon diameter and myelination thickness, i.e., from the fiber diameter and the g-ratio. Based on our g-ratio results (g = 0.54) and the median outer fiber diameters (1.19 µm for CC and 0.94 µm for SWM), we estimated conduction velocities using Rushton's formula [20], as generalized (equation 3) by Schmidt and Knösche [35]. According to this model, we estimated a substantially higher median conduction velocity in CC (median 3.4 m/s) than in SWM (median 2.7 m/s).

## Discussion

### Differences in axon diameter and myelination between SWM and CC

We provide robust evidence supporting the hypothesis that, in human white matter, short association fibers have smaller axonal radii and are less myelinated than long fibers. Our results are based on a statistical analysis of a large dataset containing approximately 400,000 myelinated axons, sampled from two SWM and six CC subregions in three human donor brains per group (with one donor contributing to both groups, totaling $n = 5$) and reveal an approximately 25% higher mean axon and outer fiber diameter in CC compared to SWM (Fig 3a and 3b). The only other available study systematically comparing SWM and CC fibers with electron microscopy was conducted in fetal sheep brain; it similarly reported a higher

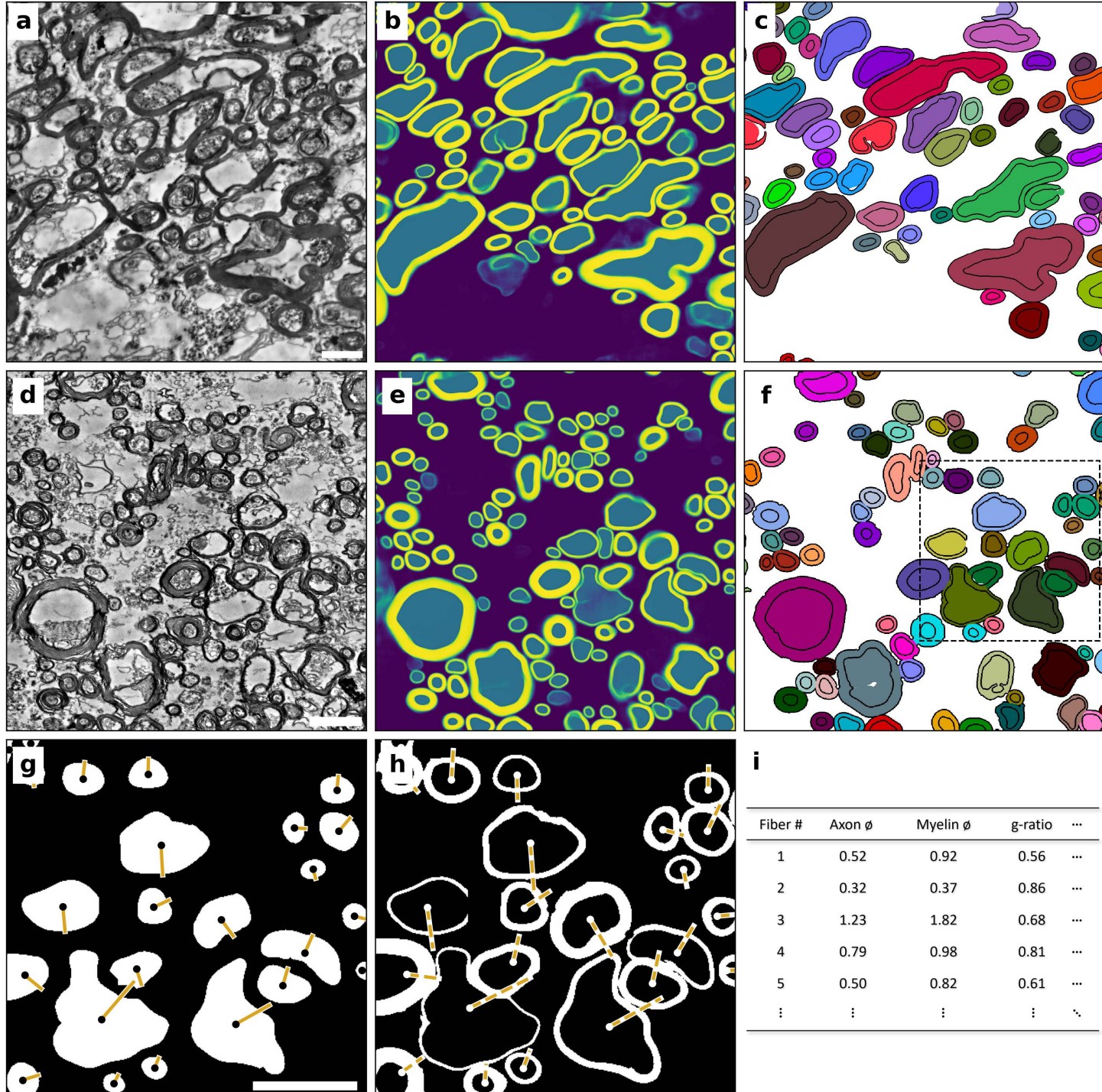

**Fig 2. Overview of the segmentation approach for electron microscopic data. a**, Preprocessed transmission electron microscopic data of human corpus callosum (CC). **b**, Semantic prediction of the preprocessed data by the trained DenseNet, returning a likelihood of each pixel to belong to either background (violet), axon (teal), or myelin (yellow). **c**, Post-processed final segmentation. Each instance of a fiber (axon and myelin sheath) is labelled with a different random color. d–f, Analogous to a–c, with transmission electron microscopic data of human superficial white matter (SWM). The dashed region in f is shown in g,h. The measurement of axon (g) and myelination (h) diameters is based on ellipses fitted to each structure. Points show the centroid of each structure, axes show half of the minor axis of the fitted ellipse for the axon (solid line) and myelin (dashed line). i, Measurements (SWM:

$n = 220,431$, CC: $n = 163,133$) from g, h are used for statistical assessment of local ultrastructure. We treat the short axis (half of which is shown in yellow in g, h) of the fitted ellipse as the actual diameter of each structure. All scale bars are 2 μm. Data available in S2 Data.

proportion of thin fibers in SWM and no difference in g-ratio [31]. Interestingly, the study also describes a higher proportion (8%) of thick fibers (>8.5 μm) in SWM, which could not be confirmed for the human brain in our study.

In the CC, a comparable light microscopic study found slightly larger axon diameters [36]. However, the authors report the radius of the area-equivalent circle instead of the short axis of an ellipse fitted to each axon, which likely explains the difference to the data presented here (see S6a and S6c Fig). The relationship between axon diameter and g-ratio in the CC (Fig 3d) corresponds well with data from the human medullary pyramidal tract, which also consists of mostly long and thick fibers specialized for long-distance information transfer [37].

Both our CC and SWM data, as well as the findings by Keyserlingk and Schramm [37], show a large proportion of larger fibers clustering around a g-ratio of 0.6, suggesting this might be an optimal ratio for long-range signal transmission in the CNS. This is also consistent with a very recent 3D electron microscopic study demonstrating similar axon diameters and myelination thickness in human SWM [30]. In contrast, we observe a larger fraction of fibers with a g-ratio >0.7 in SWM (Fig 3e). Therefore, we believe that SWM likely contains a greater variety of different fiber types than the CC, resulting in a larger variance of g-ratios.

### Regional variability within SWM and CC

While we treat CC and SWM as single groups in this study, it is very likely that there are intra-regional variations of axonal properties within subregions of each group. In the CC, Aboitiz and colleagues [32] and Mordhorst and colleagues [38] describe axon diameters in subregions of the CC as decreasing from the genu to the posterior midbody, and then increasing from the midbody to the splenium. In contrast, Riise and Pakkenberg [39] report a linear upward trend of axon diameters from anterior to posterior CC regions. Therefore, further studies examining local variations within the CC are required.

In the SWM, fiber characteristics may also vary between different locations and specialized subregions. For example, Yoshino and colleagues [40] describe two distinct SWM regions in ferrets, where fibers with smaller diameters are located very close (<50 μm) to the cortical border, highlighting local variability of the SWM. Furthermore, they propose that U-fibers are more myelinated than deep white matter fibers [41]. We could not confirm these findings in our human brain samples.

A case of potential local white matter fiber plasticity in human brains exists in the primary motor cortex (M1). Each finger, wrist, and elbow is represented twice in human M1: once in area 4a and once in area 4p [42,43]. The representation in 4p is closely linked to the somatosensory perception area and is preferentially active during combined sensory and motor tasks, whereas area 4a is hypothesized to be connected more closely to the corticospinal tract and associative frontal areas [44–46]. Therefore, if shorter SWM fibers are indeed thinner and less myelinated than longer SWM fibers, SWM beneath 4a might contain thicker and more myelinated fibers than 4p, where a larger proportion of fibers project to neighboring somatosensory areas. Similarly, in the visual cortex, color-sensitive and ocular dominance columns in the primary visual cortex V1 have distinct connectivity patterns to thick, thin, and pale stripes within V2, potentially reflecting different structural properties among these connecting fiber populations [14].

### Impact of fiber structure on conduction velocity

The conduction velocity of a fiber is determined by several factors, including its axon diameter and myelination thickness. A well-tuned conduction velocity is crucial for coordinating neural activity across different brain regions through white matter fibers. Short association fibers, such as those directly connecting the motor and somatosensory cortex, are approximately 4 cm long. For such a fiber, with a median conduction velocity of 2.7 m/s (see Fig 3f), the time needed to conduct

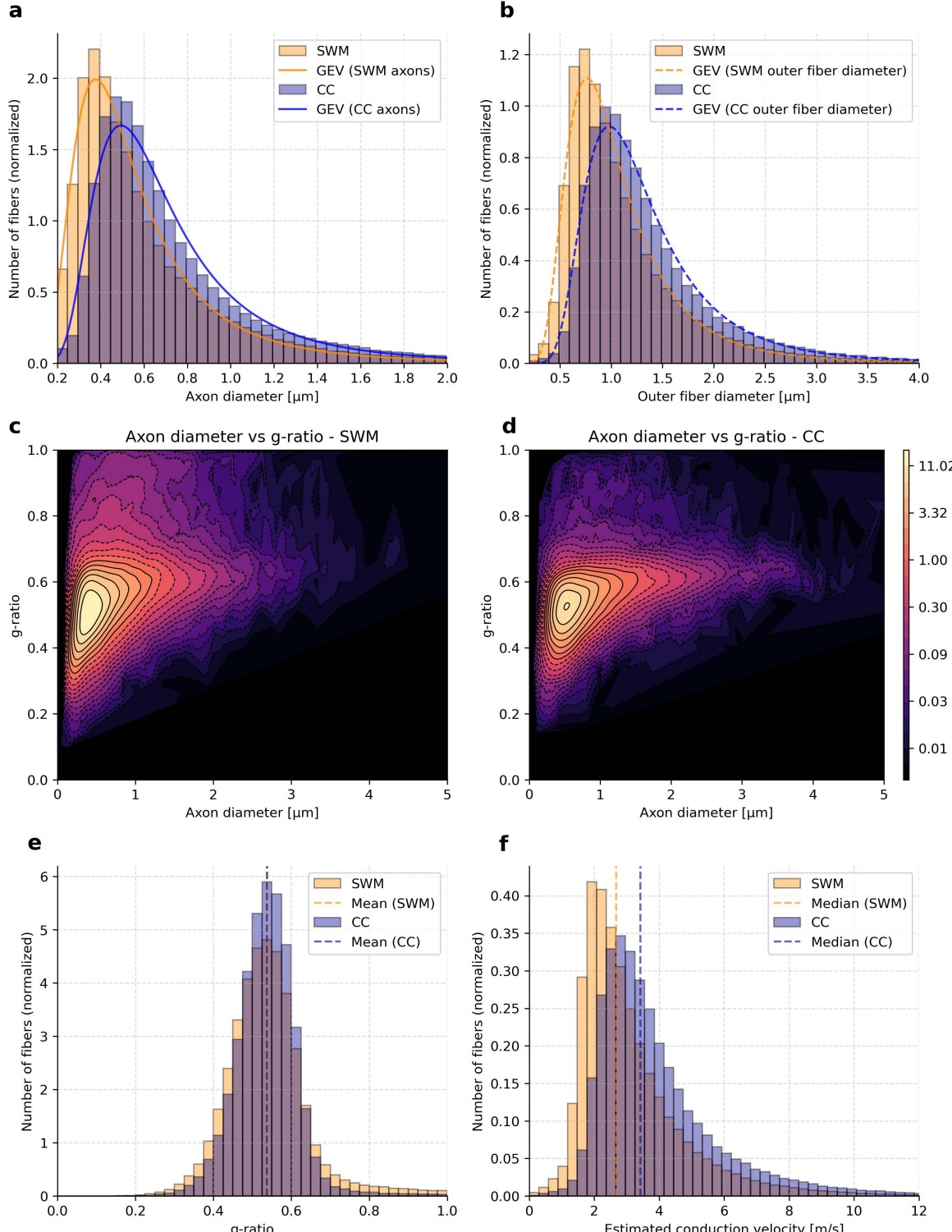

**Fig 3. Longer white matter tracts show higher axon diameters, thicker myelination, higher estimated conduction velocities, but no difference in mean g-ratio. a, b,** Distributions of superficial white matter (SWM) and corpus callosum (CC) axon diameters (a) and outer fiber diameters (b) with fitted Generalized extreme value (GEV) functions. Parameters for each GEV are given in Fig 4b. **c, d,** Densities of measured axon diameter vs. g-ratio in SWM (**c**) and CC (**d**). Contour levels are estimated from Gaussian kernel density in log scale. **e,** Histogram of g-ratio distribution in CC and SWM.

SWM mean is 0.538, CC mean is 0.537. **f**, Estimated distribution of conduction velocities of CC and SWM fibers, based on the formula by Rushton [20], modified by Schmidt and Knösche [35]. The median connection velocities are 3.4 m/s (CC) and 2.7 m/s (SWM). Data available in S3 Data.

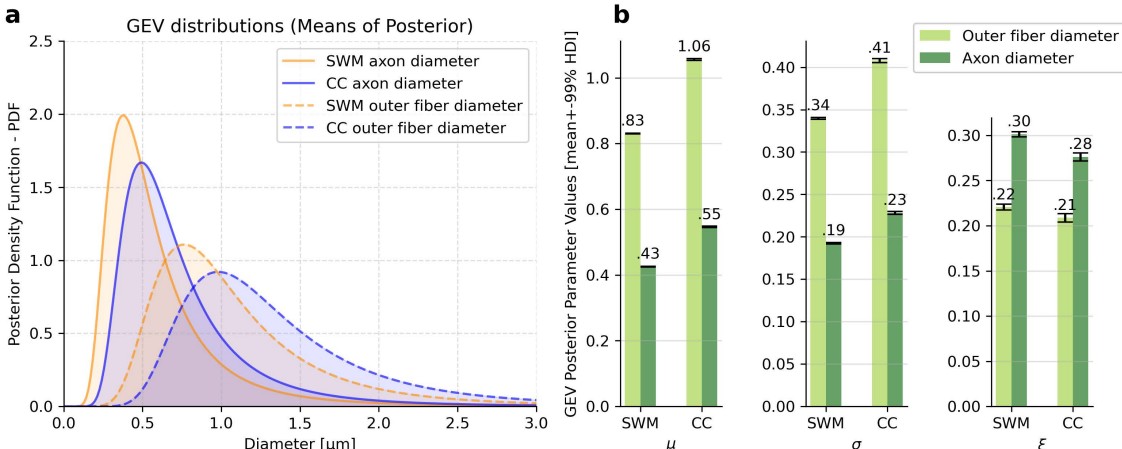

**Fig 4. Modeling axon diameter and myelination thickness with generalized extreme value (GEV) functions.** a, GEV functions fitted on superficial white matter (SWM) and CC axon diameter and outer fiber diameter distributions. Histograms for the underlying data are given in Fig 3a and 3b. b, Parameters of GEV functions fit to CC and SWM data. Error bars display 99% highest posterior density interval range. Displayed parameters are $\mu$ (location parameter), $\sigma$ (scale parameter), and $\xi$ (shape or 'tailedness' parameter). Data available in S3 Data.

a signal from motor to somatosensory cortex would be about 14.8 ms. Putting this into perspective with the typical chemical synaptic delay (~1 ms), this shows that precise tuning of conduction velocity is essential for synchronizing temporally coded neural activity in different parts of the brain.

## Optimized fiber structure is crucial for efficient information transfer under biological constraints

A key optimization challenge in large brains with a significantly expanded neocortex is managing the vastly increased number of possible connections between neuronal somata. Counterintuitively, despite the approximately 1000-fold greater number of neurons in the human brain compared to the mouse brain, human neurons do not receive more synapses on average [47]. This presents a complex problem regarding white matter fiber organization: which connections should be maintained and prioritized in terms of space and energy, and which should be reduced in caliber or eliminated altogether? Addressing this requires a finely tuned balance between axon diameter, myelination thickness, packing density, metabolic cost, and conduction velocity for each fiber, while maintaining circuit function through precisely coordinated timings [48]. In line with this notion, several studies show that 'wiring costs' required to connect the cortical areas are a relevant factor for brain organization, in addition to the hierarchical structure of the cortical areas [49,50].

Our results are in line with these observations. We show that in SWM and CC fibers, both axon diameter and outer fiber diameter scale with the fiber length—longer fibers tend to have larger diameters. In both groups, axon and outer fiber diameters scale proportionally to each other, maintaining a stable g-ratio across distances. We propose that the physical distance between connected brain regions directly influences the diameter of the connecting white matter fibers: as distance increases, fewer but larger fibers are utilized to efficiently transmit signals across longer tracts. This is consistent with previous observations in model animals (e.g., Chen and colleagues [48]). Finally, this implies that long-range connections rely on increasingly sparse but high-capacity transmission via fewer, larger-diameter, and therefore faster-conducting fibers.

## Technical considerations for sampling human material

It is difficult to gather high-quality electron microscopic data from human donors due to frequently prolonged *post-mortem* delay (20–40 h) before fixation and because fixative perfusion is not standard practice in a clinical pathology setting (as discussed in Morawski and colleagues [51]). Due to these issues, axons and myelin sheaths in electron microscopic data are frequently damaged, or, in the case of axons, degraded (Fig 2a and 2d). This makes reliable and automated analysis of these data challenging. It is unknown whether this degradation affects all fibers equally, regardless of axon diameter and myelin thickness. Therefore, a systematic bias related to the speed of degradation cannot be excluded. Furthermore, only brains of elderly individuals (60–78 years) were used for this study, which likely influences the myelination state of white matter fibers [52,53].

## Analytical considerations for two-dimensional transmission electron microscopic data

A matter of discussion in studies regarding myelinated fibers is the correct way of using a two-dimensional image to determine the diameter of a three-dimensional tubular structure, which is likely not perfectly orthogonal to the cutting plane of a two-dimensional image. In our case, we fitted ellipses on each structure, analogous to the morphometric measurements employed, e.g., by Abdollahzadeh [54]. We believe that this approach has two major advantages over alternative strategies: (i) When cutting a tubular structure at an angle, the intersection with the cutting plane creates an ellipse. The minor axis of this ellipse (= the shortest distance between two sides of the ellipse passing through the center) is always equal to the true diameter of the three-dimensional tubular structure. This is especially useful in our case, as we expect a large amount of non-orthogonally cut fibers in the SWM. (ii) Additionally, when structures are damaged (e.g., parts of myelin sheath missing or partly broken down), fitting an ellipse allows a reasonable estimate of the structure before degradation. A downside of this approach is that we cannot differentiate between (a) round, but oblique cut fibers (which this method would accurately measure) and (b) oval, but orthogonally cut fibers (where this method would underestimate the diameter of a structure). Similarly, to the fiber diameter, a variety of mathematical definitions of the g-ratio exist [55]. The simplest definition, which we follow in this manuscript, is that the g-ratio is equal to the axon radius divided by the total fiber radius, including its myelin sheath. As discussed above, one can; however, use different underlying morphological features of an image to contribute to this value. Alternatively, the axon radius can be inferred, e.g., from the radius of the circle with an equivalent radius (as performed by, e.g., Behanova and colleagues [56]) in three-dimensional electron microscopic data). Due to the data in this manuscript being two-dimensional sections with varying degrees of diagonally cut fibers (potentially more in SWM than CC), we refrain from employing an analogous approach as it would overestimate the size of diagonally cut fibers (see S6a and S6c Fig), and therefore likely skew the comparison between SWM and CC. An alternative method for estimating myelin thickness relies on skeletonizing the distance-transformed segmented structures, and yields similar results as fitting an ellipse and using the minor axis (see S6b and S6d Fig).

The strength of the current study is the very large number of analyzed fibers, which was enabled by the use of deep learning algorithms for image analysis. These approaches may entail certain limitations. Due to the 'black-box' functionality of deep learning approaches, we cannot exclude possible systematic biases in the automated analysis of our electron microscopic data. For example, due to an imbalance in the amount of training data from CC and SWM regions, we might more reliably segment fibers in SWM that are more similar to CC fibers. While our validation efforts did not show concerning biases (see S1–S6 Figs), it is possible that there are hidden effects. Furthermore, we 'idealized' obvious artifacts in the training data (e.g., partly broken-down myelin sheaths), which is not typical and might contribute to systematic errors. If there are hidden effects, they are likely to be amplified for the measurement of the g-ratio, as the inaccuracy of two measurements (axon + outer fiber diameter) is multiplied in the process. The mean absolute error for the g-ratio (not including false positives and negatives) is 0.023 (CC) and 0.057 (SWM), see S5a and S5b Fig.

## Conclusions

In this study, we show a robust difference in white matter fiber composition of human SWM and CC. However, a variety of open questions remain. It is unclear how variable white matter fiber structure is underneath neighboring cortical areas with different functions (e.g., primary motor areas 4a and 4p). Furthermore, there is still conflicting evidence about the differences of ultrastructure across CC subregions [32,39]. We are currently investigating both of these questions. Another open question is a precise model of how myelination and fiber thickness is regulated and which factors contribute to the plasticity of white matter fibers. The presented methodological approach allows for analyzing large populations of fibers. It is therefore the method of choice to answer the above-mentioned questions.

The displayed data indicate that white matter fiber structural variation between short and long fibers is present on the level of axon diameters and myelination thickness, but not in the (mean) g-ratio. However, we find a higher portion of high g-ratio (>0.7) fibers in SWM than in CC. We hypothesize that this is due to a higher variety of different signaling distances present in SWM, which is manifested in a more diverse fiber composition. We argue that the axon and myelination thickness of a fiber results from a balance of complex interactions between various factors: The fiber's length, the physiological demands of its associated white matter tract, the tissue's physical constraints, and the locally available energy budget.

We assume that the measured $g$-ratio of 0.54 reflects a biologically optimized myelination state in both groups of white matter fibers. Therefore, we also derive the exponent alpha in equation (3) from our data by reversing Rushton's original reasoning for the optimal $g$-ratio: $a = -\ln(g)$. Based on our data, we obtain $a = 0.62$, which differs only slightly from Schmidt and Knösche's result of $a = 0.68$ [35] and results in a 3% higher estimate of the conduction velocity.

With this study, we present the first large-scale analysis of white matter fiber ultrastructure in the human brain, proving a long-standing hypothesis in the field. While previous studies either relied on indirect measurement through MRI-based methods or on tissue from model animals with vastly different brain sizes (and therefore also vastly different connection distances), here we provide a comprehensive description of structural specialization in long and short white matter tracts. We show that fibers in short white matter tracts are thinner, less myelinated, but have the same mean g-ratio as fibers in longer tracts. Furthermore, we highlight how these structural specializations are integral to the correct tuning of signal conduction velocities. For future connectome studies in the human brain, we argue that it is essential to include structural information such as these, or information such as conduction velocity derived from these measures. White matter fiber structure is finetuned to achieve optimal conduction velocity, energy efficiency, efficient volumetric packing of fibers, and precise conduction of temporally coded signals in the human brain.

## Methods

### Sourcing of human tissue samples

The brain samples used in this study (5 cases, 4 male, aged 61 (*post-mortem* delay: 20 h), 74 (*post-mortem* delay: 24 h), 78 (*post- mortem* delay: 40 h), 74 (*post-mortem* delay: 24 h), 60 (female, *post-mortem* delay: 22 h) years) were provided by the Paul Flechsig Institute—Center for Neuropathology and Brain Research at the Leipzig University Medical Faculty, the Department of Neuropathology Leipzig, and the Institute of Forensic Medicine at the University Medical Center Hamburg-Eppendorf, following ethical guidelines and with proper permissions. The entire procedure of case recruitment, acquisition of the patient's personal data, the protocols and the informed written consent forms of all participants or the relatives of the deceased patients, performing the autopsy, and handling the autopsy material have been approved by the responsible authorities (approval by the Ethics Committee of the University of Leipzig; approval #205/17-ek and approval #WF-74/16, by the Ethics Committee of the University Clinic Hamburg-Eppendorf). The study was conducted in accordance with the principles expressed in the Declaration of Helsinki. Neuropathological assessment revealed no signs of neurological diseases. All samples were immersion-fixed at 4 °C in a solution of 3% formaldehyde and 1% glutaraldehyde in PBS buffer at pH 7.4 for at least 4 weeks before sectioning.

## Preparation of human brain tissue samples

Samples were taken from SWM below primary motor, somatosensory, and primary and secondary visual cortex (see Fig 1). For the motor- and somatosensory cortex, we identified the hand knob and the corresponding somatosensory region that is on the posterior side of the central sulcus. A subset of the data from SWM was cut orthogonal to the main fiber direction (U-shape perpendicular to the sulcus), and another subset was cut in parallel to the main fiber direction to check for a potential bias with respect to the cutting angle (S7 Fig).

CC samples were obtained from six regions (rostrum, genu, anterior midbody, midbody, isthmus, splenium) of the corpus callosum. CC samples were cut orthogonally to the long axis of the CC.

For electron microscopy, 70-μm thick vibratome sections were cut. At the same time, we also cut slices for histology, requiring 40 μm sections. The tissue block was embedded in low-melting agarose (2% agarose, heated, cooled down to 37 °C, and set around the tissue block on a cold metal plate). Once solid, excess agarose was trimmed. The stage of the vibratome (HM 650 V, Thermo Scientific) was coated with superglue, the block placed on top, and weighed down. After the glue set, the stage was installed in the vibratome, with the surrounding bath filled with PBS. The cutting process involved two steps: a coarse trim of 125–250 μm, then fine cuts for the 70 μm sections for electron microscopy and 40 μm for histology. The cutting speed was 1–1.4 mm/s, with a frequency of 70 Hz and a knife amplitude of 0.8 mm. The thicker sections for electron microscopy were stored in 0.2M cacodylate buffer pH 7.4, while the thinner histology sections were kept in PBS + 0.1% sodium azide.

## Nissl staining

Vibratome sections of 40 μm thickness were used for Nissl staining. Sections were mounted on glass slides and air dried for 10 min at 40 °C, then rehydrated and rinsed in distilled water for 3 min. The sections were then moved through a graded series of ethanol (70%/85%/95% ethanol for 5 min each), and then kept in 95% ethanol for 30 min. Then, they are moved through a descending alcohol series (95%/85%/70% ethanol) for 5 min each. Following this, the sections were thoroughly rinsed with distilled water several times.

The samples were stained for 10 min with an aqueous solution of 0.1% cresyl violet and quickly rinsed with distilled water. The sections then underwent differentiation through a series of ethanol concentrations (70%, 85%, two rounds of 95%, and finally 100%) for approximately 2–5 min under visual control. After achieving the desired level of differentiation, the sections were rinsed twice with toluene and then mounted with Entellan in toluene under a fume hood.

## Gallyas silver staining

The Gallyas silver impregnation technique was used to visualize myelinated fibers in brain tissue. The staining was performed on 40 μm vibratome sections. Initially, sections were incubated in a mixture of pyridine and acetic anhydride (2:1) for 30 min, followed by three washes in distilled water (5 min each). Contrast staining was achieved by immersing the sections in ammonium silver nitrate solution (1% ammonium silver nitrate, 1% silver nitrate, 3 ml/L sodium hydroxide (4%)) for 30 min. The reaction was stopped by washing the sections three times with acetic acid (0.5%), for 3 min each.

For the development step, three different solutions are prepared. Fifty ml of solution A (5% anhydrous sodium carbonate in $H_2O$) is mixed with 15 ml of solution B (0.2% ammonium nitrate, 0.2% silver nitrate, 1% tungstosilicic acid) and 35 ml of solution C (same composition as solution B, with an additional 7.3 ml/L formaldehyde [37%]). Additionally, 160 μl of bleaching solution (5% each of TETENAL Colortec RA-4 BX PART 1 and TETENAL Colortec RA-4 P BX PART 2 in $H_2O$) was added. After bleaching, differentiation was performed with sodium thiosulfate (0.5%) for approximately 10 min under visual control. After a final 10-min rinse in running tap water and a brief wash in distilled water, sections were dehydrated in an ascending ethanol/$H_2O$ series (70%, 85%, 95%, 100%) for 5 min each, then twice for 5 min each in toluene. Finally, the slides are mounted in Entellan for long-term preservation.

## Transmission electron microscopy—Sample preparation

For transmission electron microscopic imaging, 70 µm thick vibratome sections were prepared using a Microm HM 650 V (Thermo Scientific, Walldorf, Germany). Small sections of SWM or CC were cut out and stored at 4 °C in 0.2 M cacodylate buffer (4.28% Na-cacodylate, 1.2% 1N HCl) pH 7.4 until further processing. For electron microscopic imaging, we used large two-dimensional fields of views (ca. 80 µm × 800 µm) and sampled tissue for each acquisition from locations spaced several millimeters apart in order to decrease the likelihood of including individual fibers multiple times within the dataset. The sections were contrasted in 1% osmium tetroxide in cacodylate buffer for 1 hour at room temperature on a laboratory shaker (Heidolph Rotamax, 70 rpm), followed by two rinses in cacodylate buffer, each for at least 1 h, and then stored overnight at 4 °C. Dehydration was carried out through a series of acetone solutions (30% for 15 min, 50% for 30 min, 70% for rinsing) with a final contrast enhancement step using 1% uranyl acetate in 70% acetone for 45 min, protected from light, and rinsed in 90% acetone for 30 min on a shaker (70 rpm) and then two times 100% anhydrous acetone for 30 min each. The samples were embedded in Durcupan araldite resin (Sigma-Aldrich Chemie GmbH, Steinheim) by sandwiching them between polyethylene terephthalate foils on glass slides. Once cured, the backing foil was removed for bonding on a Durcupan block, and the remaining foil was peeled away. Initial 0.5 µm semithin sections were stained with toluidine blue and imaged using the AxioScan Z1 slide scanner for orientation. For large field of view transmission electron microscopy, 50 nm sections were prepared with a Leica Reichert Ultracut S and mounted on slot grids (SF 162 N5 Provac, type $1 \times 2\,mm^2$) coated with a 10–20 nm formvar film.

## Transmission electron microscopy—Imaging

Two-dimensional digital electron micrographs were acquired at 80 kV using the LEO 912 EM OMEGA (Carl Zeiss, Oberkochen, Germany) equipped with an on-axis YAG scintillator and a 2k × 2k CCD 16-bit camera with ImageSP software (TRS, Moorenweis, Germany). Large area field of views (80 µm × 800 µm) were achieved by automatic tile scanning with 10% overlap, with a pixel size of 4.32 × 4.32 nm. The resulting images were stitched using ZEISS ZEN (version 3.9). The stitched images were then manually cropped to exclude border areas of the imaging slide with no tissue on it and physically damaged areas of the tissue.

## Transmission electron microscopy analysis—Segmentation

We labelled ~16,000 two-dimensional sections of myelinated fibers as training data and trained a DenseNet (see Huang and colleagues [34]; Ronneberger and colleagues [57]) using Uni-EM [58]. In brief, we trained a network of 10 dense blocks and 10 dense layers, using the square loss function. All other parameters are given in the accompanying repository (options.json). For training and inference, we downsampled the raw data by the factor of 16 (×4 per axis) and applied a local contrast enhancement (CLAHE, Contrast Limited Adaptive Histogram Equalization). Prediction was conducted via the inference interface of Uni-EM, using a receptive field of 2048×2048 pixels. For the post-processing after prediction, the semantic segmentation mask (Fig 2b) was thresholded and corrected with binary operations and filtered for unrealistic fibers (e.g., g-ratio >1, unrealistically large or small fibers).

All custom python code is available at github.com/PhilipRuthig/ShortvsLongfibers/. In brief, we preprocessed the data with a CLAHE filter to enhance local contrast (Fig 2a and 2d). These pictures were then predicted by the trained DenseNet to yield a semantic segmentation (Fig 2b and 2e). This semantic segmentation is thresholded to generate binary masks for (a) axons and (b) myelin. In order to separate 'kissing' myelin sheaths, these masks are joined together, and a distance transformation is performed. The inverse of this distance transform is used to perform a watershed method, with seeds at every peak of the distance transform, generating a unique identifier for each axon+myelin pair. After labeling every fiber with a unique identifier, the fiber is split back to its axon and myelin portions using the binary masks generated before (Fig 2c and 2f).

Due to the sample conditions (aged human *post-mortem* brain tissue), fibers are frequently damaged. Therefore, we decided to not directly measure the segmented binary structures, but measure fitted ellipses to each structure with the same normalized second moment. This solution ensures that even damaged structures (e.g., sections of myelin sheath missing) are estimated reasonably well. In addition, this solves the problem of oblique cuts: The minor axis of an elliptical cut of a tubular structure is identical to the true diameter of a tubular structure, irrespective of the cutting angle. The measurements taken from each structure (i.e., ellipse fitted on a structure) include: minor and major axis, eccentricity, orientation, and g-ratio. For this study, we defined the g-ratio as the ratio between the minor axis of the ellipse fitted on the axon divided by the minor axis of the ellipse fitted on the myelination, as these are the most reliable measures according to our validation (see S1 and S2 Figs).

All custom python scripts used in this work are based primarily on numpy [59], scikit-image [60], and scipy [61]. All plots were generated with Matplotlib [62] and Pylustrator [63].

## Transmission electron microscopy analysis—Validation

For the validation of our segmentation pipeline, we trained a separate validation model. For the training of this validation model, we excluded a set of randomly chosen images from the training data (both from CC and SWM), and validated the predictions of the validation model against the manually generated labels for these excluded images. The segmentation pipeline reached IoU scores of 0.89/0.83 for axons and myelination in CC data, and 0.78/0.54 for SWM data. We also quantified Dice coefficients, which were 0.94/0.91 for CC (axons/myelination), and 0.77/0.70 (axons/myelination). We also quantified the 95th percentile Hausdorff distance over the validation data (ca. 150 μm × 150 μm) for CC (0.04 μm for axons, 0.06 μm for myelination) and SWM (0.38 μm for axons, 0.41 μm for myelin).

Additionally, to these per-pixel validations, we performed validations on the instance (i.e., single fiber) level, comparing the manually labeled (ground truth) structures to the respective predicted structures of the validation model. Firstly, we compared population-level differences of the manually labeled data against the prediction of our validation model (S1 and S2 Figs). To check if specific fibers are distorted in a certain way in the prediction compared to the manually labeled data, we matched each predicted fiber with its corresponding manually labeled fiber in the validation data (fibers with >40% overlapping areas are treated as the same fiber). All fibers that had no corresponding fiber (<40% overlap) in the prediction or validation data were treated as false positives or negatives, respectively. The measurements taken from matched predicted and ground truth fibers were plotted against each other in a paired way (S3 and S4 Figs). In each of these plots, a perfect prediction would mean all points would be on the plotted diagonal (or horizontal, in Bland-Altman plots), and a relatively tilted linear correlation would mean that the given measurement is biased in the respective direction. In addition to displaying the performance on this linear scale, we created Bland-Altman plots [64,65], as a method to assess agreement between two measurement techniques (in our case: manual labeling and automated segmentation). To check if the error of the main outcome measures (axon diameter, outer fiber diameter, g-ratio) is correlated with fiber size, we also plotted the individual error dependent on axon diameter in S5 Fig.

To investigate whether oblique cuts influence the measurements of our method, we also added a control for cutting angle to our data. We cut approximately half of our SWM samples in parallel to the main fiber direction (U-shape perpendicular to the sulcus), as described by blunt dissections (e.g., Catani and colleagues [1]), and the other half orthogonally to the main fiber direction. We show the resulting axon and outer fiber diameters and corresponding histological sections in S7 Fig.

## Bayesian modeling

For the description of each measured axon and outer fiber diameter distribution, we chose to fit a Generalized Extreme Value distribution (GEV, $G(x)$). It is defined by a location parameter $\mu$, scale parameter $\sigma$, and a shape ('tailedness') parameter $\xi$ according to Coles [66]).

$$G(x) = \frac{1}{\sigma} t^{\xi+1} \cdot e^{-t}$$
$$\text{with}$$
$$t = \left(1 + \xi \frac{x-\mu}{\sigma}\right)^{-1\xi}$$

(1)

For the Bayesian statistics used in this study, we used PyMC version 5.10.0 and PyMC experimental 0.0.15. MCMC sampling details are given in the associated repository (https://github.com/PhilipRuthig/ShortvsLongfibers, mcmc_CCvsSWM_GEV.ipynb). Modeling-related visualizations (S10–S12 Figs) were generated using Arviz 0.16.1 [67]. For prior information, we chose the same weakly informative priors (based on a GEV fit to the manually labelled training data for our deep learning model, see S10 Fig) for SWM and CC data. The models were designed analogous to the PyMC case study "Generalized Extreme Value Distribution" by Colin Caprani [68] using PyMC [69].

### Estimation of the conduction velocity

The velocity of an action potential along a myelinated axon is determined by the distance between two consecutive nodes of Ranvier and the time it takes for the action potential at the next node to reach threshold and trigger a new spike (time-to-spike). The node distance is the sum of the internode length ($L$) and the node length ($l$). The time-to-spike depends on fiber geometry (axonal diameter, myelin thickness, and node area) and on the physical properties of a fiber (e.g., capacitance, resistance). As a result of evolutionary optimization, the internode length is determined by the outer fiber diameter $D$ and g-ratio [20], which can alternatively be expressed as a function of $d$:

$$L \propto Dg(-ln\,g)^{0.5} = d(-ln\,g)^{0.5}$$

(2)

It is generally accepted that, for large axons, velocity—assuming an optimized g-ratio—scales with internode length and thus with fiber diameter [20,35]. In small axons, however, the shorter internode length increases the relative impact of node length on the overall conduction velocity [35,70]. As a result, the relationship between conduction velocity and fiber diameter becomes slightly supralinear. Schmidt and Knösche proposed a generalized form of Rushton's relationship.

$$v \propto d(-ln\,g)^{\alpha}$$

(3)

Here, the exponent alpha, which derives to 0.5 from the classical cable equation for a myelinated segment, depends on the internode-to-node length ratio $L/l$ with an increasing sensitivity for $L/l$ ratios below 100. In their paper, Schmidt and Knösche reported an empirically optimized value of alpha = 0.68 that best matched their simulation results. Since this equation only returns relative conduction velocities as a function of axon diameter and g-ratio, we extracted a scaling factor of 7.5 µs⁻¹ from displayed data in Schmidt and Knösche [35] Fig 5 to estimate absolute conduction velocities. In their study, Schmidt and Knösche presented five scenarios within their theoretical approach to describe action potential propagation. These scenarios gave different dependencies of velocity and outer fiber diameter. Looking for the most reasonable one, they also provided fitted parameters to data obtained by a biophysical model which was presented in Arancibia-Cárcamo and colleagues ([70] 'Scenario D (AC)'). This biophysical model includes internode and node length. Schmidt and Knösche found this to best match their results with the conclusion to present the generalized form of Rushton's relationship introducing the exponent alpha. Additionally, they found alpha to be dependent on the ratio internode length to node length.

Since our data are generated from human donor samples, where direct measurements of internode and node lengths are difficult to obtain, we performed a plausibility check using the H01 dataset released by Shapson-Coe and colleagues [71]. We identified two small-caliber SWM axons which had $L/l$ ratios below 50. Their internode lengths were 87 and 108 µm, and the four corresponding node lengths ranged from 1.5 to 4.9 µm. These values fall within the range covered by

scenario D (AC) and correspond to conduction velocities of approximately 2.5 m/s. This is consistent with our results and supports our *g*-ratio-based estimation of the conduction velocity.

## Supporting information

**S1 Fig. Comparison of manually labeled and automatically segmented measures on a validation subset (CC).** This plot shows population-level differences in the training data (manual label) and our prediction on the same image. The image used was not used for the training of the validation model. The comparison shows only minor systematic biases. All plots show mean±sd (blue) and median (orange) of area, major and minor axis length and eccentricity of CC axons (**a**), CC outer fiber measures (**b**). Data available in S5 Data (https://zenodo.org/records/15720452, [72]).
(PNG)

**S2 Fig. Comparison of manually labeled and automatically segmented measures on a validation subset (SWM).** This plot shows population-level differences in the training data (manual label) and our prediction on the same image. The image used was not used for the training of the validation model. The comparison shows only minor systematic biases. All plots show mean±sd (blue) and median (orange) of area, major, and minor axis length and eccentricity of SWM axons (**a**), SWM outer fiber measures (**b**). Data available in S4 Data (https://zenodo.org/records/15720452, [72]).
(PNG)

**S3 Fig. Instance-paired data of leave-one-out validation of segmentation algorithm in CC.** In addition to population-level measures (Figs 1 and 2), we quantified individual biases by matching each structure in the predicted data to the corresponding structure in the validation data (more than 40% overlap with the corresponding validation structure). Plotted blue diagonal (left) and dashed red (right) lines show theoretical perfect segmentation without any bias. Plots show the correlation of predicted measures of each structure with the validation data (left column) and Bland-Altman plots (right column) of CC axon diameter (**a,b**), CC outer fiber diameter (**c,d**), CC g-ratio (**e,f**). The Bland-altman plots show that most points are within limits of agreement, indicating good agreement between manually labeled data and the automated prediction. Data available in S5 Data (https://zenodo.org/records/15720452, [72]).
(PNG)

**S4 Fig. Instance-paired data of leave-one-out validation of segmentation algorithm in SWM.** In addition to population-level measures (Figs 1 and 2)), we quantified individual biases by matching each structure in the predicted data to the corresponding structure in the validation data (more than 40% overlap with the corresponding validation structure). Plotted blue diagonal (left) and dashed red (right) lines show theoretical perfect segmentation without any bias. Plots show the correlation of predicted measures of each structure with the validation data (left column) and Bland-Altman plots (right column) of SWM axon diameter (**a,b**), SWM outer fiber diameter (**c,d**), SWM g-ratio (**e,f**). The Bland-altman plots show that most points are within limits of agreement, indicating good agreement between manually labeled data and the automated prediction. Data available in S4 Data (https://zenodo.org/records/15720452, [72]).
(PNG)

**S5 Fig. Instance-wise measurement error for paired data, measured against axon diameter.** These plots show the axon diameter of the fiber plotted against the measurement error (delta of validation − prediction value) of that structure to show potential biases in particular groups of small or large fibers. These paired error values are shown for g-ratio in CC (**a**) and SWM (**b**), axon diameter in CC (**c**) and SWM (**d**), and outer fiber diameter in CC (**e**) and SWM (**f**). In all cases, the quantified error is far lower than the quantified difference between CC and SWM. Data available in S4 and S5 Data (https://zenodo.org/records/15720452, [72]).
(PNG)

**S6 Fig. Comparison of morphological measurement methods.** Scatter plots show individually paired results of two different methods for determination of axon diameter (**a,c**) and outer fiber diameter (**b,d**) in SWM (**a,b**) and CC (**c,d**). One method (plotted on the *x* axes) generated the results by fitting ellipses to the segmented structures and measuring the minor axis of the fitted ellipse, and the other method (plotted on the *y* axes) utilized a method analogous to the morphometry analysis conducted in Behanova and colleagues [56]. In brief, the axon diameter was measured by calculating the diameter of the equivalent area circle, and the myelin thickness was measured by distance transformation of the masked myelin. Both were added together for **b** and **d**. As expected, measuring the area of the equivalent circle for axon diameter increases the measured areas, due to oblique cuts not being corrected for (CC: +27%, SWM: +48%). Examining the difference between these methods on a data that is not instance-matched (**e,f**), we also find that axon diameter is overestimated when measuring the radius of the area-equivalent circle compared to measurements taken on the minor axis of the fitted ellipse. In contrast, the outer fiber diameter of measuring the fitted ellipse is almost identical to the outer fiber diameter estimated using the equivalent circle radius and skeletonizing myelin analogous to Behanova and colleagues [56]. Data available in S4 and S5 Data (https://zenodo.org/records/15720452, [72]).
(PNG)

**S7 Fig. Raw axon and outer fiber diameters for parallel and orthogonally cut SWM.** ( a) shows an exemplary fiber stained slice of human SWM (gallyas silver staining), with cutouts showing (b) parallel and (c) orthogonally cut sections of SWM. The arrow indicates the main fiber direction. (b,c) show toluidine blue stained sections, with b showing more longitudinally/diagonally cut fibers than c. (d,e) Axon diameters (d) and outer fiber diameters (e) of SWM data, split according to cutting angle. Measurements were taken from TEM data, as described in Methods. Data available in S3 Data (https://zenodo.org/records/15720452, [72]).
(PNG)

**S8 Fig. Eccentricity of fitted ellipses within the different subgroups.** This plot shows the distribution of eccentricity of the ellipses that were fitted on structures (**a**: axons, **b**: myelin), and their means in four different subgroups: Both sampled regions (CC and SWM) and the measured structures. Means are 0.73 (CC axons), 0.72 (SWM Axons), 0.67 (CC Myelin), 0.68 (SWM Myelin). Data available in S3 Data (https://zenodo.org/records/15720452, [72]).
(PNG)

**S9 Fig. Histograms of measured axon and outer fiber diameters of different biological samples.** Each distribution shows the measured axon (**a,c**) or outer fiber diameter (**b,d**) in the CC (**a,b**) or SWM (**c,d**), respectively. Data available in S3 Data (https://zenodo.org/records/15720452, [72]).
(PNG)

**S10 Fig. Prior predictive sampling for fitting a GEV distribution to CC and SWM data.** (a–d) show prior predictive GEVs for SWM axon diameter (a), SWM outer fiber diameter (b), CC axon diameter (c), and CC outer fiber diameter (d). (e) Histograms of 200 Prior samples for each of the parameters. Data is generated from prior assumptions.
(PNG)

**S11 Fig. Sample traceplots from MCMC sampling.** Left column shows histograms of all posterior samples for a set of four parameters. Different line styles represent 4 different chains. Right column shows trace plots of the MCMC sampling procedure for a single chain each. Data is not directly available due to data size limitations, but can be generated from running the MCMC script included in the github repository (mcmc_CCvsSWM_GEV.ipynb).
(PNG)

**S12 Fig. Posterior predictive sampling.** For each group of structures (CC and SWM, axon and outer fiber diameter), 150 randomly drawn posterior samples are drawn and plotted here (blue), alongside their mean (yellow) and the observed data (black). **(a)** shows CC axon diameter, **(b)** shows CC outer fiber diameter, **(c)** shows SWM axon diameter, **(d)** shows

SWM outer fiber diameter. Data is not directly available due to data size limitations, but can be generated from running the MCMC script included in the github repository (mcmc_CCvsSWM_GEV.ipynb).
(PNG)

## Acknowledgments

We sincerely thank Laurin Mordhorst, Torsten Bullmann, Thomas Knösche, Maëlig Chauvel, Gesine Müller, and Nico Scherf for helpful discussions regarding the analyses conducted in this manuscript. We also thank Ruth Stassart for her comments on earlier versions of the manuscript. Fig 1a was created using biorender.com (https://BioRender.com/0vdautg).

## Author contributions

**Conceptualization:** Siawoosh Mohammadi, Nikolaus Weiskopf, Evgeniya Kirilina, Markus Morawski.

**Data curation:** Philip Ruthig, David Edler von der Planitz, Tilo Reinert, Evgeniya Kirilina.

**Formal analysis:** Philip Ruthig, David Edler von der Planitz, Maria Morozova, Tilo Reinert.

**Funding acquisition:** Siawoosh Mohammadi, Nikolaus Weiskopf, Evgeniya Kirilina, Markus Morawski.

**Investigation:** Philip Ruthig, David Edler von der Planitz, Maria Morozova, Katja Reimann, Carsten Jäger, Markus Morawski.

**Methodology:** Philip Ruthig, Markus Morawski.

**Project administration:** Markus Morawski.

**Resources:** Markus Morawski.

**Software:** Philip Ruthig.

**Supervision:** Nikolaus Weiskopf, Markus Morawski.

**Validation:** Philip Ruthig, David Edler von der Planitz, Maria Morozova, Siawoosh Mohammadi.

**Visualization:** Philip Ruthig, David Edler von der Planitz, Markus Morawski.

**Writing – original draft:** Philip Ruthig, David Edler von der Planitz.

**Writing – review & editing:** Maria Morozova, Katja Reimann, Carsten Jäger, Tilo Reinert, Siawoosh Mohammadi, Nikolaus Weiskopf, Evgeniya Kirilina, Markus Morawski.

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
