## [Editor Report · Decision Letter 0]

16 Oct 2024

Dear Dr Morawski, 

Thank you for submitting your manuscript entitled "Human short association fibers are thinner and less myelinated than long fibers" for consideration as a Discovery Report by PLOS Biology.

Your manuscript has now been evaluated by the PLOS Biology editorial staff as well as by an academic editor with relevant expertise and I am writing to let you know that we would like to send your submission out for external peer review.

Once your full submission is complete, your paper will undergo a series of checks in preparation for peer review. After your manuscript has passed the checks it will be sent out for review. To provide the metadata for your submission, please Login to Editorial Manager (https://www.editorialmanager.com/pbiology) within two working days, i.e. by Oct 18 2024 11:59PM.

Kind regards,

Christian

Christian Schnell, PhD

Senior Editor

PLOS Biology

cschnell@plos.org

---

## [Decision Letter · Decision Letter 1]

17 Dec 2024

Dear Dr Morawski,

Thank you for your patience while your manuscript "Human short association fibers are thinner and less myelinated than long fibers" was peer-reviewed at PLOS Biology. It has now been evaluated by the PLOS Biology editors, an Academic Editor with relevant expertise, and by several independent reviewers. 

In light of the reviews, which you will find at the end of this email, we would like to invite you to revise the work to thoroughly address the reviewers' reports.

As you will see below, the reviewers agree that the study is well executed and provides important insights. However, they mention some methodological concerns that need to be addressed with additional analyses and explanations in the manuscript, for example that the control for the parallel cuts is unclear and there seems to be no control for diagonal cuts. Reviewer 3 also mentions an important statistical concern (#4). 

Given the extent of revision needed, we cannot make a decision about publication until we have seen the revised manuscript and your response to the reviewers' comments. Your revised manuscript is likely to be sent for further evaluation by all or a subset of the reviewers.

**IMPORTANT - SUBMITTING YOUR REVISION**

*Re-submission Checklist*

*Published Peer Review*

*PLOS Data Policy*

*Blot and Gel Data Policy*

Sincerely,

Christian

Christian Schnell, PhD

Senior Editor

PLOS Biology

cschnell@plos.org

REVIEWS:

Reviewer #1: The authors present a study on the electron microscopic evaluation of differences in fiber diameter and myelin sheath thickness in callosal and short white matter fibers. The found thicker and more highly myelinated long range as compared to the short range fibers, based on an automated deep learning based segmentation of the electron microscopy data from human brain tissue samples in different subcortical white matter regions underneath primary motor and sensory regions and corpus callosum from 5 postmortem human brains.

The study addresses a very relevant and timely topic which provides a profound basis for connectomic analyses and understanding of differences in information transfer across brain regions.

I nevertheless have several things which shall be addressed:

* Methods: The authors mention that they labelled 'about 16,000 cells' manually. Could the authors expand a bit more on what is meant by that? As the authors obviously did not label the cell bodies, but the axons, how could they assure that they labelled 16,000 different cells? And as this cannot mean that they labelled the whole axon of each cell, how did they assure that they did not label the same axon, and thus, same cell in consecutive sections?

* Methods: The authors state in their section about validation that "we trained a validation model on a subset of randomly chosen labeled training images and validated its prediction against the last image not inside the subset." - what exactly is meant here with 'the last image not inside the subset' if the subset was chosen randomly from the whole amount of images?

* The authors mention several validation analyses for excluding biasing effects in the analysis and data in the main text, referring to the supplemental figures. It would be helpful to the reader if these additional analyses are explained somewhere similar to a 'main' analysis as the figures are hardly understandable as they stand now.

* Results / Discussion: The authors explain differences in their results in comparison to a previous study as potentially resulting from differences in measuring axon diameter (ellipse vs. circle). While this is certainly a valid potential explanation, one might wonder if the circle of the ellipse is the more reliable way of estimating a fiber's diameter, given the distortions in postmortem tissue vs. the real-life situation. To allow for direct comparison and estimation of the delta between these approaches, it might be worthwhile if the authors provide both measures.

* Results: As one of the major results of the study, the authors emphasize the analysis on conduction velocity for which they now demonstrated differences between different brain regions with different types of fibers. While this is generally true, it needs to be mentioned that this result is mainly derived from a calculation based on an existing formula. There is no direct prove of differences in conduction velocity, though. Thus, I would not necessarily call this a direct result of the present study, but more of a possibility that could now be implemented in studies on conduction velocity.

* Discussion: As one of the limitations, the authors stress the issue with working with human tissue and with artefacts arising from the processing of this tissue. As they used tissue from five body donors, it would be really helpful to see some comparative examples of tissue snippets and how they were processed with the automatic segmentation algorithm to have a better idea of the variability within the data and about the methodical variability.

Reviewer #2: In this paper, the authors assess the axon diameter and the myelination of long fibers in the corpus callosum and shorter fibers in superficial white matter. The axon diameter and myelin thickness of short fibers in human brains is currently not well described and this understanding is of high value to the field. The study assesses the axon diameter and myelin thickness in long fibers in the corpus callosum, and short fibers between sensory and motor areas and between V1 and V2. They used transmission electron microscopy in 5 older post-mortem brains and trained a U-net to evaluate 400.00 fibers. They find that the longer fibers are thicker and the shorter fibers are thinner, as is expected. They also assess the g-ratio and find that the g-ratio is comparable across the two fiber types with a value of ~0.6. This has important implications for understanding brain structure and function given the variable conduction velocities that these different fiber types can have.

Major:

As there could be potential bias of the U-net towards larger fibers, the performance of the U-net on the smaller fibers should be shown in the same was as in the example in Figure 2 (which is from corpus callosum data). 

How is the diameter calculated for fibers that have an ellipsoid shape, like many examples in Figure 2? What are the potential explanations for the ellipsoid shape?

The control for the parallel cuts is unclear and there seems to be no control for diagonal cuts. For the parallel cuts, figure S6 should show these data, and states that: "Two example slices for the parallel cut can be seen in Fig. S1 c,d. Orthogonal cuts are not shown." However, no such slices are shown. For diagonal cuts, it seems like a relatively diagonal slice could increase the estimated diameter while having a relatively minor effect on myelin thickness, but this does not seem to be the case for Figure S6. Is it possible to explain this? If a diagonal cut consistently leads to increased diameter estimates, this could systematically bias the results.

Minor:

It would be helpful if the abstract stated the method used for the measurements. 

Figure 1C is missing a scale bar.

The variability between subjects should be shown.

Was there and variability between V1-V2 and sensory-motor fibers?

Reviewer #3 (Jussi Tohka): The manuscript describes a computational pipeline and experiments on transmission electron microscopy (TEM) images of the human brain. It applies a standard deep learning-based segmentation technique (U-Net semantic segmentation followed by watershed for instance segmentation) to segment axons and their myelin sheaths from TEM images of five donors. The manuscript concludes that there are substantially smaller fiber diameters and lower myelination in superficial white matter compared to the corpus callosum, while the g-ratios are nearly equal. I was tasked with evaluating the technical aspects of the manuscript.

The segmentation of 400,000 axons in human-derived data represents a comprehensive piece of work that has the potential to generate new insights into the organization of the human brain. I have a few technical questions, concerns, and discussion points, as well as some editorial suggestions (point 6). I think that the proper answer to question 4 could provide a proxy to "independent validation". Answers to each of the comments 1 - 5 are essential to ensure the technical validity and rigor of the work 

1. If I understood correctly, all the measurements were performed on 2D slices. How does this influence the results, especially considering that axons do not all align according to the imaging plane? Some discussion on this matter would benefit the manuscript. Figure S6 provides some insights, but I do not understand why orthogonal cuts are not shown.

2. The computation of the g-ratio by fitting ellipses (likely not ellipsoids) seems overly simplistic given the non-ellipse-like cross-sections of the axons. The mean myelin thickness can be easily obtained by a distance transform and compared to the square root of the cross-sectional area of the axons, as done elsewhere. Would this change the results? Specifically, are the cross-section shapes similar between superficial white matter (SWM) and the corpus callosum (CC)?

3. If I understood correctly, the evaluations in the supplementary figures were done using a leave-one-donor-out approach. Please confirm. It would be instructive to see image segmentation performance measures in addition to the Intersection over Union (IoU). I would prefer to see the average Hausdorff distance, as it is easier to interpret in this context.

4. The statistical analysis is the most concerning part of the manuscript. It appears that the manuscript aggregates the measurements from five donors without analyzing whether this is appropriate or if a hierarchical modeling approach should be used instead. There are many possibilities to demonstrate this and perform the hierarchical analysis if needed, so I leave it to the authors to decide on the best approach. Additionally, the manuscript's statements about the results are descriptive. While I do not think p-values or similar metrics are necessary, the manuscript should quantify the sizes of the differences in measures between the CC and SWM. It might also be beneficial to soften the title of the manuscript.

5. I would welcome more detail about the watershed method used after the semantic segmentation. It is unclear whether the manuscript trains a U-Net, DenseNet, or a combination thereof. Please clarify. Additionally, information about the network training, including its specific architecture, loss function, most important hyperparameters, and optimization algorithm, should be provided (this can be included in the supplement). Why was the data downsampled? What threshold was applied?

6. The manuscript suffers from some sloppy writing. For example, phrases like "White matter connects neighboring and distant cortical areas," "pixel resolution," "human samples," and "by matching instances with >40% identity" require rephrasing. I would prefer the results to be presented in the past tense, as they relate to these particular experiments, not general facts. Also, I found references to "a fundamental principle" unclear because it is not specified what principle is being referred to. I think that the distribution function of GEV has a typo in it.

---

## [Editor Report · Decision Letter 2]

4 Jun 2025

Dear Markus,

Thank you for your patience while we considered your revised manuscript "Human short association fibers are thinner and less myelinated than long fibers" for consideration as a Discovery Report at PLOS Biology. Your revised study has now been evaluated by the PLOS Biology editors and the Academic Editor.

Based on the assessment of your revision from the Academic Editor, we are pleased to offer you the opportunity to address the a few comments in a revision that we anticipate should not take you very long. We will then assess your revised manuscript and your response to the reviewers' comments with our Academic Editor aiming to avoid further rounds of peer-review, although we might need to consult with the reviewers, depending on the nature of the revisions.

* We would like to suggest a different title to improve its accessibility for our broad audience: 

Differences in diameter and myelination of human short and long cortico-cortical white matter fibers suggest a fine tuning of conduction velocities

* Please carefully revise your manuscript for language. There are many places where the language appears overly complicated or sloppy, for example in the abstract and Introduction: “A long-standing hypothesis in the field is that the longer a fiber tract is, the larger is the axon diameter and the thicker is the myelination of the local fiber population, as this allows more efficient and faster information transfer over long distances ..."

* The discussion should be organized more coherently. The first paragraphs until line 199 mix technical considerations and across-species comparisons. The following discussion of the conceptual significance of these findings is in contrast too short.

* The following paragraph (l 208 ff) on developmental/aging consideration is not well connected to the previous and subsequent section. Furthermore, the discussion then comes back to technical details concerning experimental measurements.

* Overall, we would like to encourage you to develop the conceptual part, for example by discussing optimal placement of areas (wire minimization) and the exponential decrease in connection weight with distance. We think that your results stand within this concept of spatial embedding.

* We also encourage you to use subheadings to organize the Discussion.

* Please add the links to the funding agencies in the Financial Disclosure statement in the manuscript details.

* Please include information in the Methods section whether the study has been conducted according to the principles expressed in the Declaration of Helsinki.

* Please specify whether the participants provided written or oral consent.

* DATA POLICY:

Regardless of the method selected, please ensure that you provide the individual numerical values that underlie the summary data displayed in the following figure panels as they are essential for readers to assess your analysis and to reproduce it: 4B, S1AB and S2AB.

* CODE POLICY

**IMPORTANT - SUBMITTING YOUR REVISION**

*Resubmission Checklist*

*Published Peer Review*

*PLOS Data Policy*

*Blot and Gel Data Policy*

Sincerely,

Christian

Christian Schnell, PhD

Senior Editor

PLOS Biology

cschnell@plos.org

---

## [Editor Report · Decision Letter 3]

22 Jul 2025

Dear Markus,

Thank you for the submission of your revised Discovery Report "Short-range human cortico-cortical white matter fibers have thinner axons and are less myelinated compared to long-range fibers despite a similar g-ratio" for publication in PLOS Biology. On behalf of my colleagues and the Academic Editor, Henry Kennedy, I am pleased to say that we can in principle accept your manuscript for publication, provided you address any remaining formatting and reporting issues. These will be detailed in an email you should receive within 2-3 business days from our colleagues in the journal operations team; no action is required from you until then. Please note that we will not be able to formally accept your manuscript and schedule it for publication until you have completed any requested changes. 

While you attend to those requests to come, please also address the following two requests:

* Please provide the link to zenodo repository that contains the source data in the Data Availability statement. 

* Please also generate a DOI for the github repository and provide this in the Data Availability statement too.

PRESS

Sincerely, 

Christian 

Christian Schnell, PhD

Senior Editor

PLOS Biology

cschnell@plos.org